# Self-Supervised Model Generalization using Out-of-Distribution Detection

**Matt Foutter**
Stanford University
mfoutter@stanford.edu

**Rohan Sinha**
Stanford University
rhnsinha@stanford.edu

**Somrita Banerjee**
Stanford University
somrita@stanford.edu

**Marco Pavone**
Stanford University
pavone@stanford.edu

**Abstract:** Autonomous agents increasingly rely on learned components to streamline safe and reliable decision making. However, data dissimilar to that seen in training, deemed to be Out-of-Distribution (OOD), creates undefined behavior in the output of our learned-components, which can have detrimental consequences in a safety critical setting such as autonomous satellite rendezvous. In the wild, we typically are exposed to a mix of in-and-out of distribution data where OOD inputs correspond to uncommon and unfamiliar data when a nominally competent system encounters a new situation. In this paper, we propose an architecture that detects the presence of OOD inputs in an online stream of data. The architecture then uses these OOD inputs to recognize domain invariant features between the original training and OOD domain to improve model inference. We demonstrate that our algorithm more than doubles model accuracy on the OOD domain with sparse, unlabeled OOD examples compared to a naive model without such data on shifted MNIST domains. Importantly, we also demonstrate our algorithm maintains strong accuracy on the original training domain, generalizing the model to a mix of in-and-out of distribution examples seen at deployment. Code for our experiment is available at: https://github.com/StanfordASL/CoRL_OODWorkshop_DANN-DL

**Keywords:** Out-of-Distribution, Domain Adaptation

## 1 Introduction

**Motivation**: Learning-based components are imperative in a well-developed robot autonomy stack to recognize patterns in high-dimensional data gathered from the environment. However, the performance of these learned models is sensitive to the distribution from which a particular input is drawn. Commonly, a supervised learning procedure assumes the training and test data are independent and identically distributed (i.i.d). Examples from a distinct distribution compared to the training data are deemed Out-of-Distribution (OOD). These uncommon OOD examples violate our i.i.d assumption and precipitate a performance loss in safety critical components. As an example, suppose we wish to develop a vision system to predict the position and orientation of a satellite in order to facilitate the autonomous interaction with and removal of non-cooperative resident space objects. We will train the model on labeled images as in Fig. 1a. That is, our training dataset consists of satellite images where only outer space is in the background. Then, while deploying this model to predict a satellite's pose, our vision system can encounter images as in Fig. 1b, which are unfamiliar due the presence of Earth in the image's background. Without a strategy to handle these OOD observations, the vision system will likely fail to make a reasonable prediction, which could result in an collision between satellites. In general, we cannot reasonably expect an agent to perfectly generalize to OOD

7th Conference on Robot Learning (CoRL 2023), Atlanta, USA.

data without prior context. But, we could potentially store and process OOD data that we encounter in deployment to improve our performance on a novel distribution through repeated exposure.

**Related Work**: Traditional domain adaptation [2] [3] is insufficient for this application since it assumes the distribution sampled at deployment is 1) static and 2) unique from the training distribution. Though, in reality, a deployment distribution can shift over time and will include a mix of in-and-out of distribution data where OOD data is markedly less frequent than the familiar in-distribution data. We would like to develop a system that is competent in both domains. Unfortunately, existing solutions in this space are expensive. Existing methods include flagging OOD inputs to be labeled by a human oracle and subsequently retraining the model, which quickly becomes expensive and time-consuming with high-dimensional, large scale datasets. Another potential solution is to request oracle labels on a small but diverse subset of OOD data and subsequently retrain the model [4]. But, the cost associated with oracle labels may be prohibitively large such that we cannot guarantee access to an oracle in all settings. Even further, under this approach, we may be disposing of potentially useful information in the unlabeled OOD images which we do not request a label on. For a more detailed discussion on current literature to address model generalization in the presence of domain shift, please see the Appendix section 5.1. For broad ap-

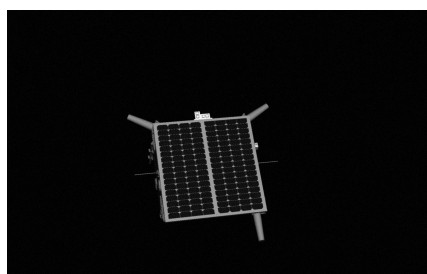

(a) In-distribution: space background

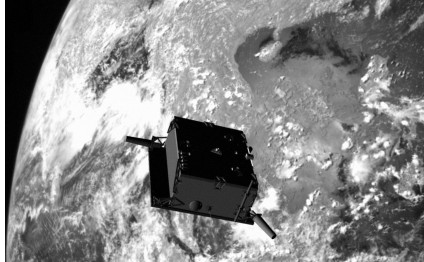

(b) OOD: Earth background

Figure 1: Distribution shift present in the SPEED dataset [1].

plicability in robotics, we desire a method to cope with OOD data that does not require an oracle for labels and learns over multiple episodes. Specifically, this project addresses how to use unlabeled, OOD examples collected over multiple episodes to improve classification on a novel target distribution without access to oracle labels. We aim to accomplish this task with sparse target examples and maintain strong classification accuracy on the source domain.

**Contributions**: This paper introduces a novel algorithm to meet the desiderata above that more than doubles target classification accuracy without target label access or drop in source accuracy relative to a naive model. Specifically, this algorithm uses an OOD detector to segment in-and-out of distribution data from an online stream of inputs. Then, this architecture learns to recognize domain invariant features between the source and target data to extend classification accuracy to both domains.

## 2  Approach

**Problem Formulation**: In this work, we consider extending a model's classification accuracy to an OOD target distribution experienced at deployment over a series of episodes. We consider the case where the source and target distribution are distinct due to covariate shift. At deployment, our model experiences a stream of unlabeled images from a mix of the source and target distribution with a heavy bias towards source representation. From this stream, we can store select inputs onboard and use them to update the model offline between deployments without access to an oracle for labeling. This formulation is motivated by applications in robotics where we deploy our robot in the wild and it experiences some uncommon and unfamiliar observations dissimilar to the training set. Ideally, after a sufficient number of episodes, the robot can begin to cope with these previously unfamiliar experiences. Through deployment, we aim to 1) improve classification accuracy on images from the target domain, 2) maintain classification accuracy on images from the source domain relative to a naive model and 3) learn with sparse OOD examples.

Table 1: DANN-DL learns with little data & no labels

| Method | Best Target Acc. | Best Source Acc. | Target labels? | Target data size |
|---|---|---|---|---|
| Oracle CNN | 0.94 | 0.98 | YES | 11,000 |
| CNN (source only) | 0.24 | 0.98 | NO | 0 |
| DANN | 0.73 | 0.95 | NO | 60,000 |
| DANN-DL w/o SCOD | 0.60 | 0.99 | NO | 150 |
| **DANN-DL + SCOD** | **0.67** | **0.99** | **NO** | $\sim \mathbf{150}$ |

**Proposed Solution**: We propose a novel algorithm, Domain Adversarial Neural Networks in the Data Lifecycle (DANN-DL), to extend classification accuracy to a previously unknown target distribution sampled at deployment. We demonstrate this algorithm's ability to increase classification accuracy in the target distribution, maintain classification accuracy in the source distribution and learn with a sparsity of target data relative to source data. This algorithm utilizes Sketching Curvature of OoD Detection (SCOD) [5] to distinguish and acquire OOD data over multiple deployments. Specifically, SCOD allows this algorithm to isolate the uncommon and unfamiliar inputs among a stream of predominantly nominal inputs during deployment. Then, at each episode with non-zero target data, the algorithm utilizes a Domain Adversarial Neural Network (DANN) [6] to recognize domain invariant features between the original training data and acquired OOD data. A more detailed discussion on the proposed approach is presented in the Appendix section 5.2.

## 3 Discussion

**Experimental Setup**: In the following results, we deploy DANN-DL on a source distribution represented by the MNIST dataset [7]. The target distribution is represented by the MNIST-M dataset [6]. Each episode consists of 75 total images of which 15 belong to the target distribution. The model begins episode zero with a random sample of 11,000 labeled MNIST source images.

We simulate deployment over a maximum of 10 episodes to ensure the total training time through deployment remains reasonably low. We train the DANN architecture with a batch size of 64 images in the source and target domain for 100 epochs with a learning rate that begins at 1e-3 and follows a cosine annealing schedule. In the Appendix section 5.3 and 5.4, we present additional results to justify the chosen framework for episodic deployment and explore halving the frequency at which we update the model at some regular rate.

**Experimental Results**: Using the experimental setup described, we simulate 10 independent trials of DANN-DL. In Table 1, we compare DANN-DL's best source and target accuracy through deployment to other relevant methods. The Oracle CNN method represents the unrealistic case where our model has perfect OOD detection and access to an oracle for labeling. With these jointly labeled datasets, the model is able to achieve upwards of $94\%$ accuracy on both distributions, which is an upper bound on performance. If we train without information in the target distribution, we achieve $24\%$ accuracy on OOD examples. Our DANN implementation modeled after [6] trains on the full MNIST & MNIST-M datasets and achieves $73\%$ accuracy on the target dataset. If we apply our DANN-DL method without SCOD, naively flagging all images at deployment as OOD, we achieve $60\%$ on the target distribution. Finally, DANN-DL using SCOD for OOD thresholding is able to achieve $67\%$ accuracy on the target with roughly $\sim 0.25\%$ of the full dataset, and in doing so, sacrifice no accuracy in the original source. In the Appendix section 5.4, we demonstrate continued deployment of DANN-DL beyond a 10 episode limit allows the target accuracy to reach comparable levels to our DANN implementation in Table 1.

The takeaways from this table are two fold. First, DANN-DL + SCOD is able to learn nearly identical target accuracy to the DANN implementation, which uses 60,000 images, with only $\sim 0.25\%$ of the data. Second, DANN-DL + SCOD's target accuracy ($67\%$) is more than double that of a naive CNN ($24\%$), is $7\%$ better than naively accepting all deployment data as OOD, and is within $30\%$ of the best possible accuracy with oracle labels ($94\%$). This first point is especially

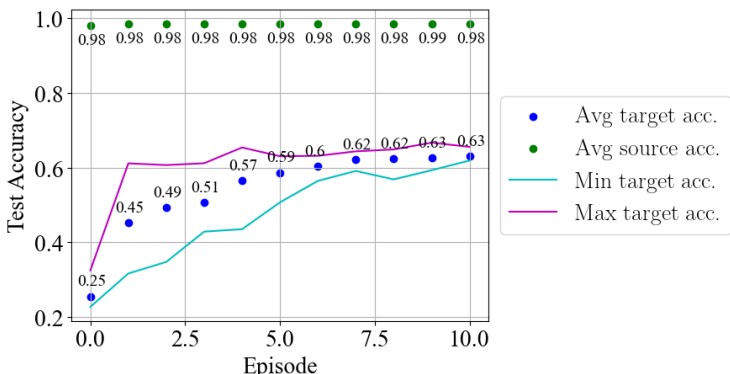

Figure 2: DANN-DL + SCOD progressively learns target and does not forget source over 10 independent trials.

important for the viability of our algorithm in a realistic robotic deployment setting where OOD data is sparse: DANN-DL is able to quickly build an understanding of the target distribution. If we have the ability to access even more OOD data, this luxury only accelerates gains in the model's accuracy, which is demonstrated in the Appendix section 5.3. The second point demonstrates the large strides in target accuracy this algorithm achieves by processing unlabeled OOD examples, in a self-supervised manner, segmented from previous deployments compared to a naive model without such OOD data. Furthermore, our results show using an OOD detector allows DANN to better capture domain invariant features between the source and target domains rather than noise in the source, increasing target accuracy by 7% compared to DANN-DL without SCOD.

In Fig. 2, we present DANN-DL's average source and target accuracy through the aforementioned 10 independent trials. This figure demonstrates the source accuracy is maintained through the data lifecycle. Additionally, Fig. 2 demonstrates the average target accuracy improves by 40% relative to a naive CNN through the data lifecycle, using only roughly 150 OOD images. Lastly, Fig. 2 demonstrates that the maximum and minimum bound on target accuracy performance converges to the average with continued deployment and data collection.

Therefore, these results demonstrate that DANN-DL improves target accuracy through deployment, by more than double a naive CNN, and in doing so, maintains source accuracy with a sparsity of unlabeled OOD examples.

## 4  Conclusion

In this work, we demonstrate a novel algorithm to leverage sparse, unlabeled OOD data to more than double target classification accuracy without a drop in source accuracy relative to a naive model. To capture a more realistic robotic deployment setting, we are currently working to develop this algorithm for satellite pose estimation and warehouse manipulation. Future research directions include potentially identifying OOD inputs in a task-aware manner. Currently, our measure of whether an input is OOD uses the functional uncertainty [8] on a naive source-only model. But, we could potentially adopt a task-aware framework that ignores traditionally OOD inputs if misclassification on the input has little to no impact on the downstream task of the system. This approach could improve the quality in OOD data collected for the specific purpose of improving downstream system-level performance.

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

## 5 Appendix

### 5.1 Detailed Discussion on Current Literature

Current literature to improve model performance on OOD inputs includes, but is not limited to, Domain Generalization (DG), Domain Adaptation (DA) and Continual Learning. Common methods in DG extend classification accuracy to an unknown distribution by learning domain invariant features among a set of labeled datasets from separate but related distributions [9], [10]. Another potential approach is in meta-learning [11] where we generalize to an artificial domain shift introduced through meta-train and meta-test batches downsampled from the original training data. However, each of these works does not utilize our ability, in most robotics settings, to sample and incorporate unlabeled data from the deployment distribution into our learning process, which makes generalization less efficient.

DA is a class of algorithms that leverages labeled training and unlabeled test data to achieve model generalization. Early work in DA re-weights the representation of training examples in the loss function such that the covariate distribution on inputs is matched between the training and test domain [2], [3]. A more modern approach to DA solves for a linear mapping to align subspaces for the training and test domain [12] from which a classifier is learned. Unfortunately, these methods transfer classification accuracy to the test distribution at the expense of accuracy on the original training distribution. Learning-based methods have demonstrated promise in extending classification accuracy to both domains by learning to recognize domain invariant features necessary for classification using a DANN [6]. But this method does demonstrate poor generalization under label shift between the training and test domains, which limits this method's application in the wild. In response, [13] augments the adversarial training pipeline with an alternative optimization to iteratively infer the test label distribution and correct for class conditional shift between domains. A reasonable estimate of the test label distribution and an adapted classifier are produced after sufficient iteration.

Even still, domain adaptation research commonly assumes the distribution sampled at test time is 1) static and 2) unique from the training distribution. However, in a robotics setting, neither of these assumptions are true. The test distribution is often a blend of in-and-out of distribution inputs. Also, the test distribution can certainly shift between, or within, multiple episodes of deployment. Therefore, in robotics, we commonly use an OOD detector to recognize when our learned components are unreliable, which could potentially be leveraged to segment in-and-out of distribution inputs over time.

Existing methods in OOD detection include, but are not limited to, utilizing runtime monitors and measures of functional uncertainty to recognize OOD examples. Recall, if our learned model operates OOD then we expect a dramatic drop in performance relative to the in distribution examples. With this result in mind, we can potentially identify when our model operates OOD by recognizing when our model has made an incorrect prediction. Specifically, [14] engineers consistency checks across comparable sensor modalities in a system to identify and diagnose the specific sub-module at fault. However, this work is incompatible with a single-component system and struggles to guarantee fault detection when the number of faults is large relative to the expression in our consistency checks. We could also use a measure of the model's own prediction uncertainty to identify OOD examples. Explicitly, [5] introduces SCOD which perturbs the weights of a trained model for a particular input and studies the degree to which the output distribution changes. This distribution change is then represented by a scalar measure of distance that we can compare to a benchmark to create an in-and-out of distribution segmentation on inputs. In [15], the authors train an ensemble of models in parallel on the same data and use the entropy of the average output distribution to give a scalar quantity compared to a benchmark for ODD detection. In each case, these methods are easy to integrate into an existing autonomy stack and enable our learned components to recognize OOD inputs that we can potentially use in a self-supervised learning process.

Another approach to achieve model generalization is continual learning. Research in continual learning iteratively updates a model with new experiences using regularization techniques to ensure inferences from past memories, or training data, are not forgotten [16][17]. Continual learning is different from traditional methods in machine learning because this algorithm class samples data from the environment over time and uses it to iteratively improve model generalization. In that way, continual learning aligns with our desire to learn a test distribution through episodic samples without forgetting classification on the original training distribution. Yet, the need for these methods to introduce a regularization strategy during parameter updates is motivated by a lack of access to past memories when onboard storage and compute power may be limited. Additionally, these methods require labels associated with the test data, which is again incompatible with our desired solution. Instead, we are interested in a setting where model training and data storage is performed offline at a central datacenter, which communicates model updates to the autonomous agent between deployment episodes.

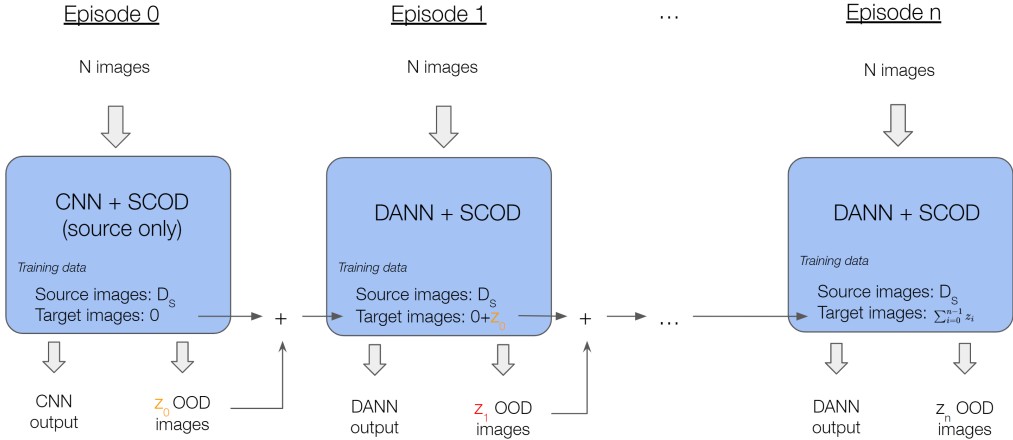

Figure 3: At episode zero, we have no apriori understanding of the target distribution and train a naive CNN to govern our agent. We use this naive CNN to form a SCOD object for OOD detection that is held constant through all episodes. Our algorithm uses SCOD to segment in-and-out of distribution data from an online stream of inputs. Then, the algorithm learns to recognize domain invariant features between the source and target data to extend classification accuracy to both domains. We continue this iteration until the target dataset is as large as the source dataset.

Current literature which operates in the setting above includes work in Diverse Subsampling using Sketching Curvature for Out-of-Distribution Detection [4]. This algorithm leverages an OOD detector to downsample likely OOD examples from an episode of inputs. The algorithm then requests oracle labels on a maximally informative subset of these OOD examples for a fixed label budget. This work is closely aligned with this project by sampling inputs from a mix of in-and-out of distribution data and by iteratively improving model generalization with the sampled OOD inputs. However, this pipeline relies on access to an oracle to label OOD inputs, which may be infeasible or expensive.

## 5.2 Detailed Discussion on Proposed Approach

The workflow for this algorithm is visually presented in Fig. 3. Across all episodes the algorithm will experience a batch of N images on which to classify. Some small fraction of these N images will come from the target distribution while the remaining images belong to the source distribution. At each episode, our algorithm has access to the original curated source dataset. In the zeroth episode, we have no a priori understanding of the target distribution. Our target dataset is empty, and therefore, we train a naive CNN based on our source dataset. We use this naive CNN to create a SCOD object to distinguish source and OOD images during deployment. In our work, the benchmark for SCOD segmentation is the 95th percentile uncertainty on the model's training data in order to achieve an expected false positive rate of roughly $5\%$ in the wild. Of note, we hold this SCOD object constant throughout the rest of deployment.

During episode zero, our CNN classifies on a stream of inputs and we use SCOD to identify OOD examples. Then, between the first and second episode, we train a DANN architecture offline from scratch to recognize domain invariant features between the source and newly augmented target dataset. We deploy the new DANN architecture in epsiode one along side the constant SCOD object. We repeat the steps described previously with a DANN architecture replacing the CNN and iterate until the target dataset is as large as the source dataset.

In summary, our task during each episode is two fold: (1) identify a subset of OOD images using our constant SCOD object and (2) classify on an episode of N images using the current episode's model architecture. In our work, we use SCOD to identify a subset of OOD images from each episode, which are added to the target dataset. We then use a DANN architecture, retrained at each episode, to generalize classification to the source and target domain. In doing so, we progressively build an understanding of the target distribution with few examples and maintain classification in the source distribution.

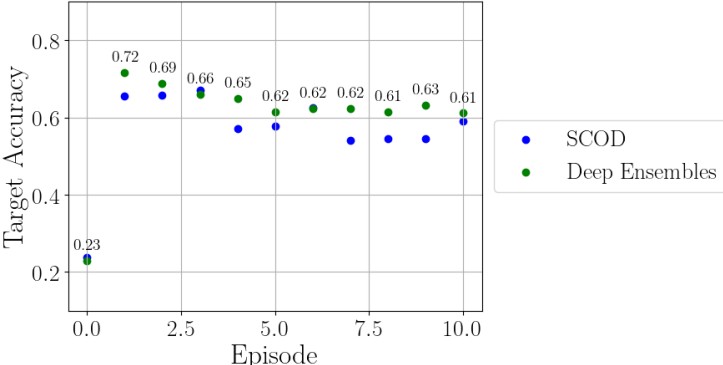

Figure 4: DANN-DL target accuracy decays with SCOD and Deep Ensembles [15] for OOD detection.

### 5.3 Episode Size and Constant SCOD

This section of the appendix will present justification for the number of images included per episode and the decision to maintain a constant SCOD wrapper through deployment. In the early stages of this project, different from the data lifecycle described previously, we exposed DANN-DL to 5,000 images per episode of which 1,000 were OOD target examples. Also, this project created a new SCOD object at each deployment episode for OOD thresholding. Initial results for this project using SCOD and Deep Ensembles [15] for OOD detection are presented in Fig. 4, which demonstrates a decay in the target accuracy through deployment. The target accuracy jumps significantly at the introduction of the first DANN architecture but continued deployment and data collection causes a decay in the target accuracy.

That is, the target accuracy saturates after the first few deployment episodes independent of the OOD detector we use. In the case of Deep Ensembles, the DANN model classifying on episode one has access to roughly 1,000 target examples from the previous episode and achieves a target accuracy of 72%, which is practically identical to the DANN implementation's accuracy introduced in Tab. 1. This result suggests 1,000 target images, at least on the MNIST-M dataset, is a diverse enough sampling of the distribution to classify well on it.

In conjunction with this dramatic increase in target accuracy comes a dramatic drop in model uncertainty on the target distribution as well. Because SCOD and Deep Ensembles perform OOD detection using the model's own prediction uncertainty, OOD detection becomes more difficult as our understanding of the target distribtution improves. In other words, creating a new SCOD object or Deep Ensembles at each deployment episode to perform the task of OOD detection is in

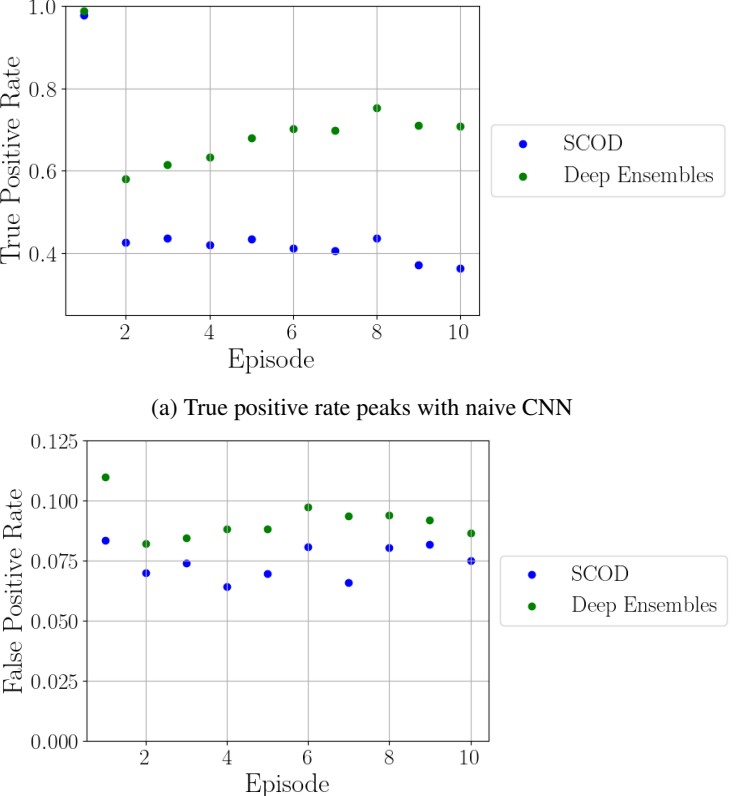

(a) True positive rate peaks with naive CNN

(b) False positive rate relatively constant through deployment.

Figure 5: DANN-DL source and target accuracy through deployment averaged over independent 10 trials.

competition with the task to improve classification accuracy on the OOD domain. This result is demonstrated in Fig. 5a where the true positive rate peaks at the first episode independent of the detection method and sharply falls in deployments thereafter. In Fig. 5b, the false positive rate remains mostly constant throughout deployment given our understanding of the source distribution is unchanged in deployment. Small changes to the false positive rate between deployment episodes are likely a product of sample bias between the 4,000 images in those episodes.

In response to these results, this project investigated exposing the model to a fewer number of OOD images per episode and holding the OOD detection object in the first episode constant throughout deployment. The results presented in the main body of this work are the best example of DANN-DL progressively learning the target distribution over mutliple episodes. Access to more OOD data per episode should accelerate growth in the model's understanding of the target distribution. This result is verified in Fig. 6 where the DANN architecture is iteratively retrained with oracle detection experiencing 64 OOD examples in Fig. 6a and 10 OOD examples in Fig. 6b per episode. In comparison to Fig. 6b, Fig. 6a learns an initially a higher target accuracy and peaks at a higher target accuracy later in deployment.

## 5.4 Halving the Training Frequency

This section of the appendix will investigate DANN-DL's source and target classification performance while halving the training frequency through deployment at some regular rate. Reducing the

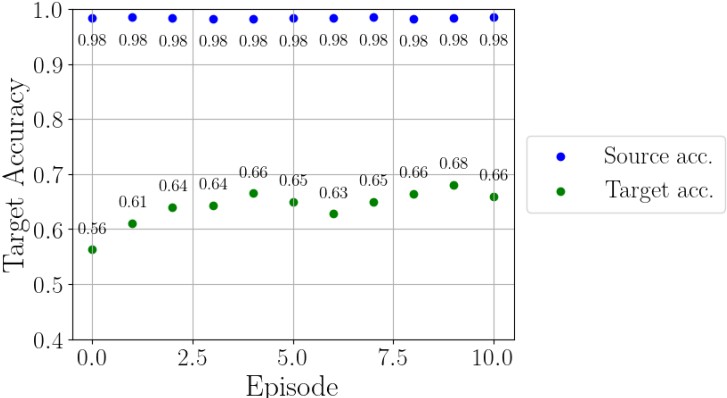

(a) DANN progressively learns with oracle OOD detection and 64 images per episode.

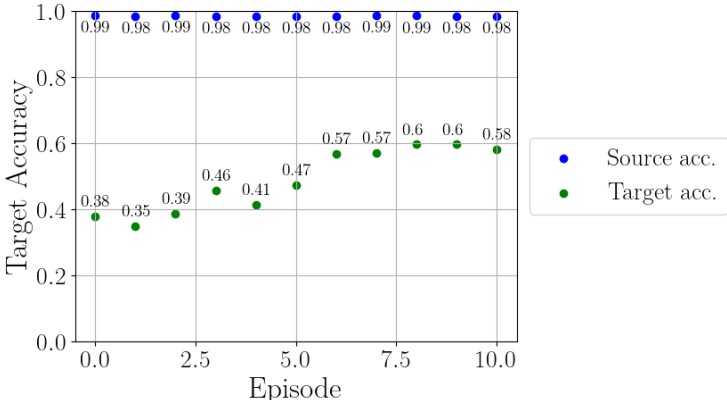

(b) DANN progressively learns with oracle OOD detection and 10 images per episode.

Figure 6: DANN-DL learns higher initial and final target accuracy using more OOD data.

rate at which we train the model is desirable as our training process can be computationally expensive and unnecessary if the model is already well adapted to both the source and target domains. In the later stages of deployment, we expect the target accuracy to improve and stabilize in which case less training may not harm performance. In this study, we choose to half the frequency of training after every 10 training cycles. That is, after the first 10 episodes, where we retrain a DANN architecture between each episode, we train a new DANN architecture once every two episodes. After another 20 episodes have passed, or 10 training cycles, we will then train a new DANN architecture once every four episodes. This iteration continues until DANN-DL achieves greater than 70% accuracy on the target domain or the sampled target dataset is as large as the labeled source dataset, whichever comes first. We chose 70% specifically as we deemed this close enough to our DANN implementation's target accuracy on the full MNIST-M dataset presented in Tab. 1.

In the following analysis, we plot the source and target accuracy with the training cycle, rather than with the episode number, to highlight the performance difference between new DANN architectures in the data lifecycle. In Fig. 7a and Fig. 7b we provide two independent trials of simulating the DANN-DL algorithm with the modified training strategy described. In each case, we approach a target accuracy of 70% long before the source and target datasets are comparable sizes. Specifically, after 15 training cycles under this setup, DANN-DL has seen 300 OOD examples. Of note, we had to train the DANN architecture for 200 epochs per cycle, else DANN-DL will not reach 70% target accuracy before the source and target datasets are comparable sizes. These results serve to justify continued investigation into a potentially modified training schedule as opposed to the constant schedule presented in the main body of this work. Also, importantly, these results demonstrate that

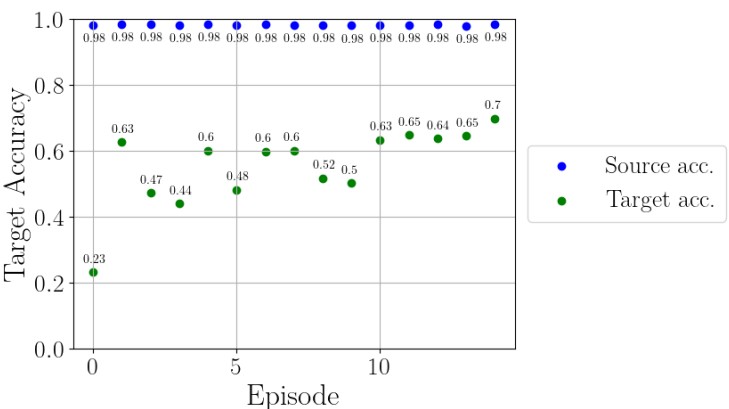

(a) DANN-DL reaches 70% after 15 training cycles.

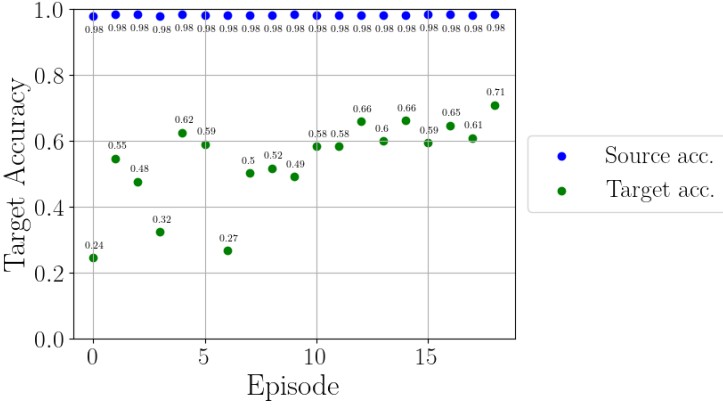

(b) DANN-DL reaches 70% after 19 training cycles.

Figure 7: Two DANN-DL trials with variable training frequency.

continued deployment our of algorithm beyond 10 episodes allows the target accuracy to approach our DANN implementation's accuracy in Tab. 1.

**Acknowledgments**

The authors would like to thank Edward Schmerling for his invaluable feedback and the reviewers for their helpful comments.

