# OpenReview forum: "Self-Supervised Model Generalization using Out-of-Distribution Detection"
_robot-learning.org/CoRL/2023/Workshop/OOD — OOD Workshop @ CoRL 2023_

### Official Review · Reviewer_XXpE · 2023-10-16
**Relevant study on data lifecycle mgmt that lacks some depth in its analysis**

**Rating:** 6
**Confidence:** 4

**Review:**

This workshop paper proposes combining an existing OOD detection algorithm with a domain adversarial NN for episodic deployment scenarios. By incorporating the OOD data after each episode, the work demonstrates that they can vastly improve the performance on the OOD data (target) while not suffering performance on the training data.

This work is very relevant to the workshop given its particular focus on one of the key occurrences of OOD data in robotics.

The writing is clear and easy to follow but is sometimes a bit poorly worded (grammatically).

To the best of my knowledge, combining OOD detection and adversarial training for the context of episodic learning is novel. However, many works do implicitly account for domain transfer (especially as the OOD detection is based purely on performance, i.e. functional uncertainty). It would be interesting to compare to recent works in domain adaptation etc.

The work could have come up with a stronger motivation on the hand of examples for the setting in which a fixed subset of data points in an episode are from a different distribution. The described setting seems more similar to settings without consequences for OOD datapoints, whereas robotics is specifically vulnerable due to real-time decision making. The typical performance degradation once an OOD data point is detected is not accounted for.

The work contains some interesting analysis in the appendix on retraining vs not modifying the OOD detector. A further analysis of this in the main contribution would have been very relevant for the community overal. Especially if it is followed by a recommendation on how to jointly improve performance and reducing the reliance on OOD detectors (which may trigger safety controllers etc.)

Overall, the results highlight that the proposed approach has its merits. As mentioned before, it lacks strong comparison methods from domain adaptation community but can be extended in the future.

---

### Official Review · Reviewer_eiYo · 2023-10-17
**Explanation of contribution and motivation needs clarity but experiments are strong**

**Rating:** 6
**Confidence:** 4

**Review:**

This work has a nice incorporation of the DANN work in terms of using representation learning as the approach to tackling OOD robustness. It makes sense that OOD robustness should be something that is achieved via adaptation. Additionally, the results look very promising.

In terms of writing, the contribution of this work as compared with DANN needs to be made more clear. Is the difference simply the incorporation of SCOD? If so, why is SCOD used instead of the DANN domain classifier? As is stated in the paper, SCOD is with respect to the source original CNN and so won’t be accurate for the evolving model unlike the DANN domain classifier.

The authors should provide more examples of problems in which the agent experiences “rare, unfamiliar observations dissimilar to the training set” instead of a complete shift (for which the unfamiliar observations would not be rare). It seems that rare triggers of the OOD detector would be due to noise rather than true distribution shift. In that case it is unclear how valuable the representation learning done by DANN would be.

The paper has strong and impressive experimental analyses, but the main contributions need more explanation.

---

### Decision · Program_Chairs · 2023-10-17

**Decision:**

Accept

**Comment:**

We agree with the reviewers’ assessment that this work is technically sound and will contribute to productive, topical discussions at the 2023 Workshop on OOD Generalization in Robotics. In particular, we appreciate that this work addresses both OOD detection and its relation to downstream OOD generalization performance, though its contributions/impact could be better established through additional discussion and baseline comparisons. We recommend the authors incorporate the reviewers’ feedback into their camera-ready submission to further improve their manuscript.